# What Do We Know about Social and Non-Social Factors Influencing the Pathway from Cognitive Health to Dementia? A Systematic Review of Reviews

**DOI:** 10.3390/brainsci12091214

**Published:** 2022-09-08

**Authors:** Marta Lenart-Bugla, Mateusz Łuc, Marcin Pawłowski, Dorota Szcześniak, Imke Seifert, Henrik Wiegelmann, Ansgar Gerhardus, Karin Wolf-Ostermann, Etiënne A. J. A. Rouwette, M. Arfan Ikram, Henry Brodaty, Yun-Hee Jeon, Jane Maddock, Anna Marseglia, René J. F. Melis, Suraj Samtani, Hui-Xin Wang, Anna-Karin Welmer, Myrra Vernooij-Dassen, Joanna Rymaszewska

**Affiliations:** 1Department of Psychiatry, Wroclaw Medical University, 50-367 Wroclaw, Poland; 2Department for Health Services Research, Institute of Public Health and Nursing Research, University of Bremen, 28359 Bremen, Germany; 3Department of Nursing Science Research, Institute of Public Health and Nursing Research, University of Bremen, 28359 Bremen, Germany; 4Institute for Management Research, Radboud University, 6525 XZ Nijmegen, The Netherlands; 5Department of Epidemiology, Erasmus MC University Medical Center, 3015 GD Rotterdam, The Netherlands; 6Centre for Healthy Brain Ageing (CHeBA), Discipline of Psychiatry and Mental Health, Faculty of Medicine and Health, University of New South Wales, Sydney, NSW 2052, Australia; 7Susan Wakil School of Nursing and Midwifery, The University of Sydney, Sydney, NSW 2006, Australia; 8MRC Unit of Lifelong Health and Ageing, Faculty of Population Health Science, University College London, London WC1E 6BT, UK; 9Division of Clinical Geriatrics, Centre for Alzheimer Research, Department of Neurobiology, Care Sciences, and Society, Karolinska Institutet, 171 77 Stockholm, Sweden; 10Radboud University Medical Center, Radboud Institute for Health Sciences, Radboudumc Alzheimer Center, 6525 XZ Nijmegen, The Netherlands; 11Aging Research Center, Department of Neurobiology, Care Sciences and Society, Karolinska Institutet and Stockholm University, 171 77 Stockholm, Sweden; 12Stress Research Institute, Stockholm University, 114 19 Stockholm, Sweden; 13Stockholm Gerontology Research Center, 113 46 Stockholm, Sweden; 14Division of Physiotherapy, Department of Neurobiology, Care Sciences and Society, Karolinska Institutet, 171 77 Stockholm, Sweden; 15Women’s Health and Allied Health Professionals Theme, Medical Unit Medical Psychology, Karolinska University Hospital, 141 86 Stockholm, Sweden

**Keywords:** dementia, cognitive function, risk factors, protective factors, social factors, cognitive decline

## Abstract

The heterogeneous and multi-factorial nature of dementia requires the consideration of all health aspects when predicting the risk of its development and planning strategies for its prevention. This systematic review of reviews provides a comprehensive synthesis of those factors associated with cognition in the context of dementia, identifying the role of social aspects and evidencing knowledge gaps in this area of research. Systematic reviews and meta-analyses from 2009–2021 were searched for within Medline, PsycINFO, CINAHL Complete, Cochrane, and Epistemonikos. Reviewers independently screened, reviewed, and assessed the records, following the PRISMA-2020 guidelines. From 314 included studies, 624 cognitive-related factors were identified, most of them risk factors (61.2%), mainly belonging to the group of ‘somatic comorbidities’ (cardiovascular disease and diabetes) and ‘genetic predispositions’. The protective factors (20%) were mainly related to lifestyle, pointing to the Mediterranean diet, regular physical activity, and cognitively stimulating activities. Social factors constituted 9.6% of all identified factors. Research on biological and medical factors dominates the reviewed literature. Greater social support and frequent contact may confer some protection against cognitive decline and dementia by delaying its onset or reducing the overall risk; however, overall, our findings are inconsistent. Further research is needed in the fields of lifestyle, psychology, social health, and the protective factors against cognitive decline and dementia.

## 1. Introduction

To date, research on dementia has provided a wide range of biological evidence, especially pointing to its genetic pathogenesis [1,2,3], as well as evidence of the effects of environmental factors [4,5]. However, increasingly, reports emphasize the importance of those modifiable factors related to the development and course of dementia [6,7]. The most frequently studied factors are cardio-metabolic and vascular disorders, smoking, low physical activity, low education, and nutritional deficiencies [8,9]. More recent studies provide findings on the importance of social aspects in the incidence of cognitive decline, indicating the protective effects of social activities, social engagement, and a supportive personal network [10,11,12,13,14].

Interest in protective mechanisms in the context of aging resulted in the term cognitive reserve (CR) [15], understood as the ability to cope with brain damage and, thus, resistance to the manifestation of symptoms of cognitive impairment. Various factors contribute to this brain resilience, thereby indirectly affecting cognitive function [16,17]. When discussing the multitude of factors determining the cognitive condition of an individual, it is impossible to ignore the interrelationships of the three areas of health—mental, physical, and social. These core domains represent a holistic and dynamic understanding of well-being [18]. The importance and interrelationship of the social aspects of health, which complement physical and mental aspects, are reflected in the concept of ‘social health’, coined to stress the influential role of one’s social and environmental resources in maintaining a balance between one’s capacities and limitations. Considering the aging population and generally facing the challenges of the 21st century, psychological and social interventions for people with dementia have gained relevance as they are based on experience and have proven to be cost-effective [18,19].

To date, few systematic reviews have been performed that capture a wide range of factors that contribute to cognitive decline. The evidence indicates a need to integrate data from various fields in order to understand the complexity of dementia and to create a framework for prevention. However, it is noted that the results mainly concern the elderly population [20] and the modifiable risk factors for cognitive decline or dementia [6,9]. A recently published study [8] has expanded the field of research on the subject by providing high-quality global evidence from 91 meta-analyses and adopting a wide scope. However, given that dementia develops many years before the first symptoms appear, it is crucial to consider a longer observation perspective to identify those factors present throughout a life span. So far, no systematic overview has been made that also includes factors related to normal cognitive functioning. From the perspective of prevention and the identification of the importance of social health, this approach can be considered a valuable complement to the existing knowledge.

Hence, this article aims to provide a comprehensive overview of the evidence regarding a variety of factors in the context of the process of cognitive decline and the development of dementia. In addition, it is novel to highlight the social aspects of this process in order to lay the groundwork for an integrated social, psychological, and biomedical framework for preventive interventions in dementia. The overview of the available knowledge allows for the identification of research gaps that may prove relevant in future studies.

Both the breadth and complexity of the issues, as well as the multitude of studies, justify the use of an overview methodology aimed at signposting the reader to relevant evidence, summarizing the existing research, or highlighting the absence of evidence [21].

The research question is: What is the current state of knowledge on those factors influencing cognitive functioning in the context of dementia development?

This overview is the first step of the European Joint Programme—Neurodegenerative Disease Research, funded by social health and reserve in the dementia patient journey (SHARED, JPND, 2019). SHARED aims to create a comprehensive framework of social health in conjunction with physical and mental health and its impact on cognitive decline and the course of dementia.

## 2. Materials and Methods

### 2.1. Search Methods

The review was prepared following the PRISMA 2020 guidelines [22] (see Appendix A for the PRISMA 2020 checklist). Two researchers (ML.-B., H.W.) independently searched five online databases: Medline, PsycINFO, CINAHL Complete, Cochrane Database of Systematic Reviews, and Epistemonikos. The search strategy was constructed in consultation with experienced researchers and combined terms related to cognitive functioning as the outcome variable, a set of ‘factor’ terms as the conditional variable, and a third group for specifying the study types, including systematic reviews and meta-analyses. The strategy combined the following keywords: (‘dementia’ OR ‘Alzheimer*’ OR ‘cognitive impairment*’ OR ‘cognitive decline’ OR ‘cognitive function*’ OR ‘cognitive reserve’ OR ‘cognitive capacity*’) AND (‘risk factor*’ OR ‘predictive factor*’ OR ‘prognostic factor*’ OR ‘course’ OR ‘cause’ OR ‘contributing factor*’ OR ‘predisposing factor*’) AND (‘systematic review’ OR ‘meta-analysis’). This strategy was adapted accordingly for all databases (Appendix A). The search was limited to human studies, the English language, journals and periodicals, and the time period: 1 January 2009–1 August 2019. An additional search for systematic reviews and meta-analyses on social factors only was conducted on 10 January 2022, with an applied publication period of 1 June 2019–31 December 2021. Bibliographies management software (Mendeley) was used to store all citations found during the entire search process and to check for duplicates. All selected publications were uploaded into PRIMARY Excel Workbook for Systematic Review for the screening process, a tool designed specifically for this purpose [23]. First, after the removal of duplicates, each record was screened by two independent reviewers (M.L.-B., M.Ł., M.P., H.W., I.S.) based on the titles and abstracts, then based on full texts against inclusion and exclusion criteria. Any disagreements were solved by a consensual discussion with a third reviewer.

### 2.2. Eligibility Criteria

Inclusion criteria were structured around the outcome, study design, the population of interest, and publication type. They included the following criteria: (1) publications reporting the relations (positive or negative) of one single factor or a number (collection) of factors on cognition or cognitive decline or dementia; (2) systematic reviews (SR) or meta-analyses (MA); (3) participants aged at least 18 years, regardless of the level of cognitive performance; (4) peer-reviewed publications, published in English. The exclusion criteria were: (1) studies on other diseases (for example with no or a weak link to dementia, or about reverse causality or with no influencing factors); (2) studies about pharmacological interventions (for example drug tests); (3) non-human studies (for example animal testing) (full exclusion criteria in Appendix A). The eligibility criteria and their clarity were checked using the Fleiss Kappa test—a chance-corrected measure of agreement between more than two reviewers on a sample of studies [24]. The interrater reliability value was 0.91. Any discrepancies were solved by consensus to achieve an acceptable level of agreement.

### 2.3. Data Extraction and Synthesis

Systematic overviews enable wide, but not deep, searches for certain topics; thus, the aim of this overview was to ‘provide a summary of existing research syntheses related to a given topic or question, not to re-synthesize, for example, with meta-analysis or meta-synthesis, the results of existing reviews or syntheses’ [25]. This paper was focused on presenting the overall frequency of the factors, without exploring the relative significance or contribution of each factor to the development or progression of dementia. 

As a starting point, the team of researchers (ML.-B., M.Ł., M.P., I.S., H.W.) screened the selected articles and models on the subject to build a framework of categories. Based on the screening, team discussions, and consultations with specialists in the field, the preliminary list of seven main categories with 59 subcategories was established. Categorization of factors was conducted during the review of the full text, and data were summarized in a prepared ‘data-charting form’ (Microsoft Excel file), containing: title, author(s), year, country, years covered, number of studies (if applicable), type of literature, type of diagnosis/health status, the age range of study participants, key findings, and 59 subgroups in the seven primary groups for categorizing reported factors. As an example, the identified factor ‘female’ was classified as the ‘sex’ subcategory in the ‘demographic’ category. The table with the data entered was checked by the first author in order to ensure uniformity, catching deficiencies and possible errors. Results were described and summarized narratively, according to three steps: (1) tabulation of the results into a predefined categorical framework; (2) the analysis of differences in the main findings between studies; (3) a synthesis of the findings under each theme and (4) a description of publication bias and heterogeneity.

### 2.4. Quality Assessment

Critical appraisal of the methodological quality of SRs and MAs is required when conducting a systematic review of reviews [25]. The authors (M.L.-B., M.Ł., M.P., D.S.) used the AMSTAR checklist [26] to independently assess each article. This tool consists of 11 items, providing a rating of domains for potential bias, and has good face and content validity. Due to the purpose of checking the quality of the methodology of the included articles, we did not plan to exclude any articles on the basis of the appraisal scores.

## 3. Results

### 3.1. Literature Search and Study Selection

The database searches of the published literature resulted in 1188 potentially relevant articles. These were supplemented with five more articles received from the SHARED consortium members at the identification stage. After the removal of 871 duplicates, titles and abstracts were double-screened in pairs, and 365 records were selected for the full-text review. Of these, 310 met the inclusion criteria and were included in this systematic overview, as shown on the PRISMA flow diagram (Figure 1). Additionally, four publications on social factors were identified through an additional search. The overview describes the results from a total of 314 records.

### 3.2. Included SRs and MAs Characteristics

The general characteristics of the included studies are shown in Table 1. An amount of 314 articles published between 2009 and 2021, ranging from 1 in 2021 to 57 in 2017, were found. The main type of study was SR (*n* = 129), and 114 articles used combined methods of SR and MA, with 71 using MAs only. Most publications were carried out within European countries (35.8%, including the UK), followed by China (28.3%), the USA (12.7%), and Australia (7.3%). The number of studies included in the SRs and MAs ranged from 2 to 291 (mean = 27.8). The age categories were created based on the provided age range of participants in the papers. The most commonly reported group was ‘mid-life and late-life’ (45.2%; 40 years and above), then ‘late-life’ (23.6%; 65 years and above), ‘people of all ages’ (13.7%; 18 years and above), and only 1% for ‘young-adulthood and mid-life’ (18–65 years). Within 52 studies, there were no descriptions of the age range of the participants. The reviewed articles most often focused on several cognitive outcome variables; the most frequently studied was dementia (any type), reported 252 times (54.6%), including Alzheimer’s disease (*n* = 160; 34.6%) and cognitive impairment or cognitive decline mentioned in 111 articles (24%). The complete characteristics of the articles can be found in Appendix A.

### 3.3. Quality Assessment Results

Of the 314 publications, 57.6% (*n* = 181) exhibited high methodological quality, with scores ranging from 8–11, 35% (*n* = 110) exhibited moderate quality (scores 4–7), and 7.3% (*n* = 23) scored ≤3 points on AMSTAR scale (see Appendix A). The domain with the least points was ‘a provided list of excluded studies’ in the selection process (14.3% of articles). ‘A priori’ scoring of the study design was based on the presence of the study objective and specified criteria for inclusion; hence, this criterion was met by 61.2% of the records. The vast majority of SRs and MAs (94%) provided a table with the characteristics of the included studies, usually in the form of a table. In 67.2% of the reviews, the scientific quality of the included studies was assessed and documented, as well as used when formulating the conclusions. There was heterogeneity in the methods used to combine the findings and an assessment of publication bias was provided in no more than 65% of the records, which suggests carefully drawing conclusions from the available data.

### 3.4. Categorization of the Obtained Data

Most of the reviewed articles reported more than one factor associated with a cognitive outcome, bringing the total number of factors found to 624. As a result of the data extraction process, the number of subcategories increased to 61 in comparison to the previously established 59 in the framework (Table 2). The seven main target categories were changed only in terms of the names to correctly reflect the content. The full list of all factors in the categorized subcategories can be found in Appendix A.

The numbers presented in Figure 2 reflect the frequency of reported factors, grouped into 7 categories and 61 subcategories, in order from most frequently reported to least frequently reported. The full references of the 314 included SRs and MAs, grouped by category, are included in Appendix A.

#### 3.4.1. Somatic Diseases and Biological Factors

Most of the factors noted were classified in the categories related to neuroscience, biology, and medicine (*n* = 275; 44.1%), marked in green, with 21 subcategories representing specific groups of diseases or branches of medicine. The most numerous subgroup was ‘somatic comorbidity (SC)’ (*n* = 138), comprising 16 clinical units: diabetes (*n* = 22), cardiovascular diseases (*n* = 21), cerebrovascular diseases (*n* = 16), hypertension (*n* = 16), obesity (*n* = 11), metabolic syndrome (*n* = 9), sleep disturbances (*n* = 8), post-operative cognitive deficits (*n* = 7), trauma (*n* = 7), inflammatory diseases of connective tissue (*n* = 5), kidney disease (*n* = 5), tooth loss (*n* = 3), hearing impairment (*n* = 2), Parkinson’s disease (*n* = 2), chronic obstructive pulmonary disease (*n* = 2), and cancer-related factors (*n* = 2). The frequency of reported factors in this subcategory is shown in Figure 3. The subgroup ‘genetic predisposition’ was second in regard to the frequency of factors (*n* = 77), followed by ‘vitamins, micro-elements’ (*n* = 16), ‘biomarkers’ (*n* = 13), ‘neuroimaging’ (*n* = 6), and ‘molecular factors’ (*n* = 3). An amount of 22 factors were grouped into the last subcategory, ‘other’, with two or fewer mentions: orthostatic hypotension, motoric cognitive risk, frailty, higher stride-time variability, hyperhomocysteinemia, bullous pemphigoid, epilepsy, conjugated equine oestrogen use, chronic periodontitis, bilateral vestibulopathy, critical illness, early appearance of extrapyramidal signs, benzodiazepine use, handgrip strength, headache, hepatic disorders, infective episodes, and sarcopenia.

#### 3.4.2. Lifestyle

The second cluster, in terms of the number of factors, was ‘lifestyle’ (*n* = 150; 24%), marked in orange, with 10 subcategories, in which 34 were related to diet, 24 to physical activity, 18 to alcohol, 17 to smoking or passive smoking, 16 to nutritional supplements, 14 to weight, 12 to cognitively stimulating activities, 9 to caffeine, 3 to sleep pattern, and 3 to ‘other’, with a single mention given to meditation, good mastication, and regular cannabis use.

#### 3.4.3. Social

In the ‘social’ category, marked in the blue color, 60 factors (9.6%) were grouped into eight subcategories: social network (*n* = 13), social contacts (*n* = 12), marital status (*n* = 9), social support (*n* = 6), satisfaction with social ties (*n* = 5), participation in social activities (*n* = 5), loneliness (*n* = 5), and social engagement (*n* = 5).

#### 3.4.4. Psychiatric & Psychological

Depression or depressive symptoms (*n* = 18), marked in the red color, were the most cited factors assigned to the ‘psychiatric & psychological’ cluster (*n* = 44; 7.1%), followed by anxiety (*n* = 9), personality traits (*n* = 3), apathy (*n* = 2), bipolar disorder (*n* = 2), delirium (*n* = 2), subjective cognitive complaints (*n* = 2), and six factors in ‘other’: psychosis, job burnout, post-traumatic stress disorder (PTSD), schizophrenia, personality disorders, and poor learning abilities.

#### 3.4.5. Socioeconomic

The majority of the factors from the ‘socioeconomic’ cluster (*n* = 40; 6.4%), marked in the grey color, were associated with education (*n* = 24), followed by the subcategories of living arrangements (*n* = 8), socioeconomic status (SES) (*n* = 5), and occupation (*n* = 3).

#### 3.4.6. Environmental

A total of 30 factors (4.8%) were designated as ‘environmental’, marked in yellow, with six subcategories: exposure to metals (*n* = 6), air pollution (*n* = 5), sun exposure (*n* = 5), exposure to pesticides (*n* = 5), exposure to low-frequency electromagnetic fields (*n* = 4), and five other factors: urban residence, rural residence, green space exposure, selenium, and solvents.

#### 3.4.7. Demographic

The least numerous category, ‘demographic’ factors (*n* = 25; 4%), marked in the pink color, included four subcategories: age (*n* = 9), race/ethnicity (*n* = 9), sex (*n* = 5), and bilingualism (*n* = 2).

### 3.5. Types of Relationships (Negative or Positive) between Factors and Cognitive Function

In 314 SRs and MAs, the authors reported those factors which provided information on the type of association with cognitive function. In 507 (81.3%) out of the 624 cases, a significant association with cognitive function was found (382 with negative and 125 with positive association). For 28 (4.5%) reports, there was no overall evidence for an association with cognitive function (no significant association). For the last 89 (14.3%) reports, this association remained unclear (most often due to inconsistent results from studies or the conclusion drawn by the authors), which made it impossible to draw definite conclusions. Figure 4 presents the percentage of reports with the identified relationship with cognitive function. 

#### 3.5.1. Negative Relationship (‘Risk Factors’)

Factors where the majority of the evidence pointed to a negative association with cognitive function were classified in the ‘somatic diseases & biological’ category (*n* = 220; 80% of all factors in this cluster), followed by ‘psychiatric & psychological’ (*n* = 39, 88.6%), ‘environmental’ (*n* = 22; 73%), and ‘demographic’ (*n* = 19; 76%). Among the most frequently studied risk factors were (Figure 5): genetic predisposition, mostly apolipoprotein E e4 carrier status (*n* = 62, with 2 reports of no association), cardiovascular disease (*n* = 21), diabetes (*n* = 18, with 4 reports of inconsistent results), depression/depressive symptoms (*n* = 18), cerebrovascular disease (*n* = 16), smoking/passive smoking (*n* = 13, with 2 reports of no association and 2 for inconsistent results), hypertension (*n* = 11, with 4 reports of inconsistent results and 1 of no association), biomarkers (*n* = 11) and deficiencies of vitamins and microelements (*n* = 10), lower education (*n* = 9), sleep disturbances (*n* = 8), older age (*n* = 8), anxiety symptoms (*n* = 8), various ethnicity (North America, North Europe, India, Sub-Saharan Africa; *n* = 7), trauma-related symptoms (*n* = 6), kidney disease (*n* = 5), inflammatory diseases of connective tissue (*n* = 5), exposure to pesticides (*n* = 5), air pollution (*n* = 5), exposure to low-frequency electromagnetic fields (*n* = 4), loneliness (*n* = 4), neuroimaging (*n* = 4), and being female (*n* = 4).

#### 3.5.2. Positive Relationship (‘Protective Factors’)

The positive relationship between a given factor and cognitive function has been reported 125 times. Over half of these cases concerned the category ‘lifestyle’ (*n* = 70, constituting 46.7% of the entire category), 20 factors were found in the ‘somatic diseases & biological’ category, 19 in ‘social’ (31.7% of the category), and 14 (35%) in ‘socioeconomic’. The most frequently mentioned protective factors belong to the following subcategories (Figure 6): diet (*n* = 22, Mediterranean diet, intake of vegetables, omega-3 fatty acids), physical activity (*n* = 17, regular activity), genetic predispositions (*n* = 12, various other than apolipoprotein E e4 carrier status), cognitively stimulating activities (*n* = 10), higher level of education (*n* = 9), frequent social contacts (*n* = 6), low to moderate intake of alcohol (*n* = 5), nutritional supplement (*n* = 4), normal BMI in midlife (*n* = 4), and caffeine consumption (*n* = 4).

#### 3.5.3. Inconsistent Results (‘Unclear Association’)

Conclusions regarding the unclear type of relationship between a given factor and cognitive function resulted from both inconsistent results from individual studies (conclusions from reviews) and discrepancies between the evidence extracted from the included SRs and MAs. Figure 7 shows those factors with inconsistent evidence but also those where there is duality (risk factor and protective factor). Among the most frequently reported are diet (*n* = 34, with 6 reports on risk factors, including high consumption of saturated fatty acids, 22 reports on protective factors including intake of vegetables, omega-3 fatty acids, 6 reports with inconsistent results), alcohol (*n* = 6, with 6 reports on risk factors, e.g., excessive alcohol intake, history of lifetime drinking or abstinence, 5 reports on protective factors as low to moderate alcohol intake, and 1 report of no association), weight (*n* = 4, with 5 reports on risk factors such as underweight or obesity in midlife, 4 reports on normal BMI being a protective factor, and 1 report of no association), caffeine (*n* = 9, with 4 reports on moderate coffee intake as a protective factor and 3 inconsistent reports on the dose of caffeine and cognitive decline association), smoking (*n* = 17, with 13 reports on risk factors, 2 reports with inconsistent results on midlife smoking, and 2 reports on no association), post-operative cognitive deficits (*n* = 4, with 2 reports of negative—cardiac operation, post-stroke delirium, 4 reports with inconsistent results, and 1 report of no association), vitamins and microelements (*n* = 16, with 10 reports on deficiencies, such as low vitamin B12 and folate as risk factors, 3 reports on vitamin C and E as protective factors, and 3 reports with inconsistent results), cancer (*n* = 2, with 1 report on breast cancer chemotherapy as a risk factor, and 1 report of a history of cancer lowering the risk of AD with 37%), exposure to metals (*n* = 6, with 3 reports on exposure to aluminium or silicon as a risk factor, 1 report on exposure to mercury as a protective factor, and 2 reports with inconsistent results on copper and iron), living environment (*n* = 8, with 4 reports on risk factors, including low SES-neighbourhood, 3 reports on protective factors such as good local community, and 1 report with inconsistent results on the association between homelessness and cognitive decline), social contacts (*n* = 12, with 5 reports on less social contacts or social isolation as risk factors, 6 reports on frequent social contacts as protective factors, and 1 report with inconsistent results), marital status (*n* = 9, with 5 reports on risk factors such as being lifelong single, widowed, or divorced, 2 reports on being married as a protective factor, 1 report on inconsistent results, and 1 report on no association).

### 3.6. Inconsistent Results on Social Factors and Their Relationship with Cognitive Functioning in the Context of Dementia Development

A total of 17 publications provided evidence on social factors. The 60 factors were grouped into eight subcategories, noting various types of association with three cognitive outcomes (Figure 8). Cognitive health reflects unimpaired cognitive functioning, which is measured in studies of cognitively healthy people, using global cognition and cognitive domains. Cognitive decline reflects general cognitive deterioration (including MCI as a prodromal stage of dementia), while dementia is a medical diagnosis of brain disease.

#### 3.6.1. Reports on Cognitive Health

A systematic review of RCTs, longitudinal, and twin studies in healthy older adults [27] showed that social activity was most consistently associated with better global cognition (improved scores), assessed using global or domain-specific cognitive measures, as well as increased brain volume. Similarly, greater social support and larger social networks were associated with better global cognition, while there was a significant relationship between social contact and better episodic memory and verbal fluency. The results by Wu et al. [28] showed that higher social engagement was a good predictor of cognitive function maintenance at a high level or a ‘minor decline’ among older adults. Cross-sectional and longitudinal studies [29] indicated that loneliness is negatively correlated with cognitive functions (specifically in the domains of general cognitive ability, intelligence quotient, processing speed, immediate recall, and delayed recall). One systematic review [30] reported no consistent results on the relationship between cognitive health and social support and social contact in older adults.

#### 3.6.2. Reports on Cognitive Decline, Dementia, and AD

A systematic review and meta-analysis of longitudinal studies [31] showed evidence of incident dementia being associated with low social participation, less frequent social contact, and loneliness but inconsistent results with social network size. The next review of longitudinal studies on cognitive decline [32] reported an association between poor social contact and loneliness, as well as a small social network, poor social support, and low satisfaction with social ties. Similarly, systematic reviews of cross-sectional studies found that social isolation is associated with an increased risk of dementia, as is loneliness with Alzheimer’s disease [29,33]. Two studies identified a larger social network as protective against cognitive decline or dementia in later life [34,35]. Conversely, a smaller social network (i.e., living alone) has been related to MCI or increased dementia risk [14,34,35,36,37]. Although one review provided evidence that satisfying contact is associated with a lower risk of dementia and a higher risk in the case of unsatisfying ties, two different SRs reported no significant association with dementia or AD [14,38]. 

Marital status was often studied for its impact on cognitive functioning. The majority of findings indicated an association between a higher risk of MCI, dementia, or AD and being lifelong single, divorced, or widowed; married participants were less likely to develop dementia [14,34,37,39]. A recent systematic review on Indigenous populations found no association between marital status and dementia but suggested that social connections may be protective against Alzheimer’s disease and related dementia [40]. Maintaining social contact was also described as a protective factor against MCI prevalence [37].

A systematic review and meta-analysis on the risk of dementia [14] emphasized the positive role of social contact, engagement, and participation in social activities in reducing the risk of dementia, and elevated risk in their absence. 

One review of longitudinal and RCTs [20] could not draw firm conclusions about the consistent association between marital status, social network, social support, and cognitive decline. Similarly, in two studies [14,38], the authors did not show a significant association between larger social network size, marital status, loneliness, strong social support, and dementia or AD.

## 4. Discussion

This paper provides a comprehensive summary of 314 systematic reviews and meta-analyses that recognize factors related to cognitive functioning and the development of dementia, with a closer look at the role of social factors and knowledge gaps in the literature.

The obtained results are in line with previous findings that show the main emphasis of dementia research is on risk factors (61.2% of 624 identified factors), while protective factors accounted for 20%. In 4.5% of cases, there was no significant association with the cognitive outcome; in the remaining 14.3%, the results were inconclusive, suggesting that these factors require further investigation. 

The majority of the studies have been grounded in the field of medicine, reporting mainly on the risk factors of dementia (80% of all factors in the ‘somatic diseases & biological’ category), providing the most evidence on somatic comorbidities (50%) and genetic predispositions (28%). The most frequent comorbidities were diabetes, cardiovascular diseases, cerebrovascular diseases, hypertension, obesity, and metabolic syndrome. This may be due to the dominant medical perspective in the field of dementia research and a bias towards treatment rather than prevention.

Applying a broad search strategy enabled the discovery of those factors related to different fields, such as health behaviors. Hence, those aspects were clustered into the ‘lifestyle’ category, where most factors (46%) were identified as ‘positively’ influencing cognition, pointing to the Mediterranean diet, regular physical activity, cognitively stimulating activities, and maintenance of normal body weight, whereas smoking, heavy alcohol use, and bad sleep patterns were associated with an increased risk of dementia. 

Due to the use of subcategories in our review, some of the results contain factors that are differently related to cognitive functioning and dementia. These are, for example, marital status (with single status recognized as a risk factor, while being married is a protective factor), or even cancer (breast cancer chemotherapy having a negative association, while one SR and MA showed that patients with a history of cancer had a 37% decreased risk of AD). Also, due to the heterogeneity of the measures used on the factors and cognitive outcomes in SRs and MAs, and given the multitude of records, it was impossible to draw unambiguous conclusions about the type of impact for most of the factors. Among them, we have found factors which need more longitudinal studies: post-operative cognitive deficits, consumption of caffeine, consumption of alcohol, or weight throughout a lifetime. However, the most consistent results are in line with previous findings on the well-established dementia risk factors—diabetes, smoking, depression, and hypertension, and the protective factors—regular physical activity and a healthy diet [6,8]. 

The frequently reported factors from other categories were level of education (a high level is related to better cognition and a low level represents a risk factor), depression or its symptoms (the majority of the evidence indicates a negative impact on cognitive health), anxiety and older age (both risk factors). In addition, the review aggregated environmental factors such as air pollution, metals exposure, pesticides, or sun exposure, most of which have been shown to have a detrimental effect on cognitive functioning.

The heterogeneous and multi-factorial nature of dementia, confirmed by our findings, requires taking all crucial aspects of health into account when predicting the risk of dementia development and also when planning prevention strategies [41].

### 4.1. Findings on Social Factors in the Context of Other Research

Social factors and their influence on cognition represents one of the key aims of this review. The findings showed many inconsistencies between the included studies. Reports indicated that low social engagement, less participation in social activities, social isolation, being single, divorced, or widowed, having unsatisfying social ties, and having a small social network can contribute to an elevated risk of dementia. This is in line with previous research showing that poor social networks and social disengagement are associated with cognitive decline and dementia by impacting physiological, psychological, and behavioral pathways [42,43,44]. The results indicate that while greater support and frequent contact may confer some protection against cognitive decline and dementia by reducing the risk or delaying the onset, the findings were not completely consistent. Although social interactions are quite often perceived as protective factors against cognitive impairment [45], we found that this has not been explored as extensively as has lifestyle and physical health factors. This contradicts recent findings [6] that social integration is one of the key modifiable factors influencing the trajectory of cognitive performance, among well-grounded factors such as physical activity, smoking, or mental stimulation. Hence, further exploration of social factors can improve the consistency and quality of this evidence and thus lead to more effective methods for intervention [46,47]. This is important as meta-analyses demonstrate that each protective factor decreases the chances of cognitive impairment, even in the presence of risk factors [9].

### 4.2. Research Gaps and Implications for Future Studies

The overview allowed for the identification of gaps in existing dementia research. First, the evidence regarding cognitive reserve and its indicators is very limited—only two records [48,49] focused on identifying CR determinants. In light of its well-established mechanism for coping with brain damage or cognitive impairments, it is an urgent area for further investigation in the context of dementia [50]. Secondly, the categorization of factors made it possible to compare areas in terms of the amount of data available. An uneven amount of evidence is observed between the relatively neglected positive aspects of psychology (e.g., self-efficacy, self-acceptance, and autonomy) and the dominant focus on mental symptoms or illnesses. Thirdly, although the category ‘lifestyle’ was the only one in which most factors were positively associated with cognitive health, associations for many factors were inconsistent. Further research that addresses the modifiable factors is still needed. Finally, as more studies focused on dementia rather than on the trajectory of cognitive decline to dementia, it is relevant to initiate longitudinal studies mapping the whole cognitive pathway in order to identify those factors which play a role in the earlier stages.

### 4.3. Strengths and Limitations

One of the key advantages of the current review was the implementation of a transparent methodology with a systematic search, selection, and analysis of the relevant literature, allowing for the replication of our findings. The scope of this review was broad; hence, the search was designed to comprehensively identify the relevant literature. For this reason, a large number of studies were selected for review, which is an advantage in summarizing evidence by combining research from various scientific fields and obtaining a wider perspective that leads to a better understanding of the topic. The synthesis of reviews and meta-analyses from the last 13 years presents the state of knowledge and current research trends. The main strength of the study was its holistic approach, identifying the role of various factors in the entire cognitive health trajectory. This is also the first such extensive review that examines the role of social aspects against other factor clusters, emphasizing an integrative understanding of health.

However, this review has several limitations. Although the inclusion criteria of SR and MA allowed for a methodology appraisal, it was not possible to compare and assess the reliability of the results obtained from individual original studies due to the heterogeneity in the methods used to combine the findings. Another limitation is the exclusion of very recently published primary studies and non-English publications. Given that social aspects are of increasing interest in dementia research worldwide, it would be relevant to include other types of publications from different countries in future reviews.

## 5. Conclusions

Connecting biological, medical, psychological, and social factors to the development and course of dementia in a novel, complex model provides opportunities for a greater understanding of cognitive decline with aging and can create a basis for its prevention and the modification of its course. Neglected research areas, such as studies on cognitive reserve and positive psychological aspects, should be targeted. The protective role of social factors in the trajectory of dementia aligns with the recent findings on social engagement being an indicator of cognitive reserve [12,13]. This confirms a need to expand research efforts around social health, which may play a buffering role, slowing down the development of the disease or reducing the degree of cognitive decline that occurs during aging.

## Figures and Tables

**Figure 1 brainsci-12-01214-f001:**
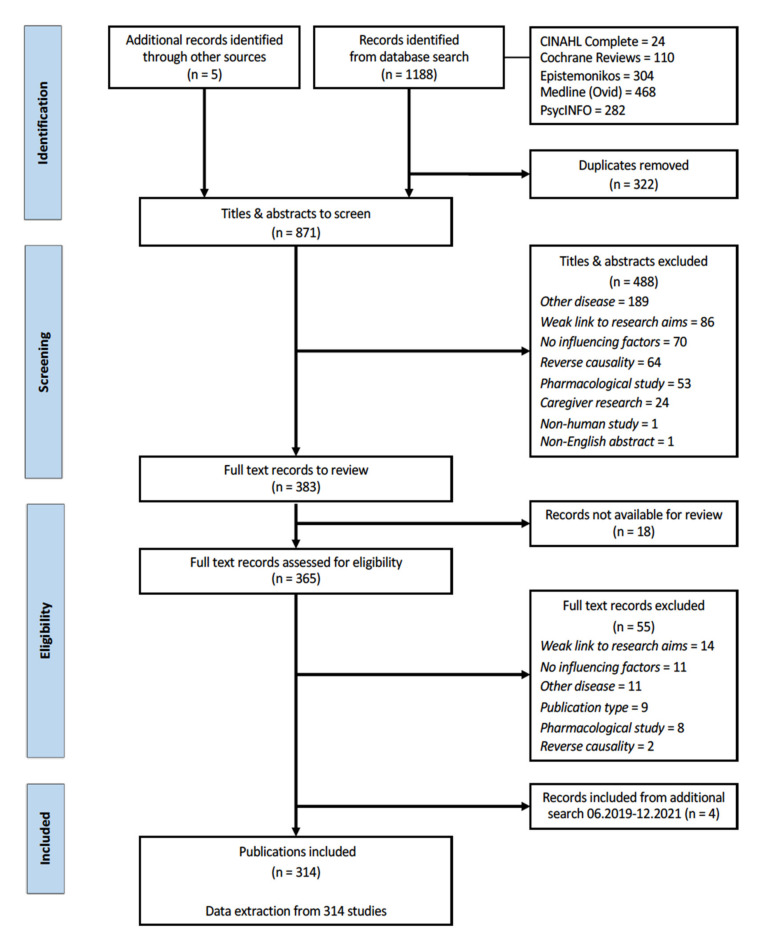
PRISMA Flow diagram.

**Figure 2 brainsci-12-01214-f002:**
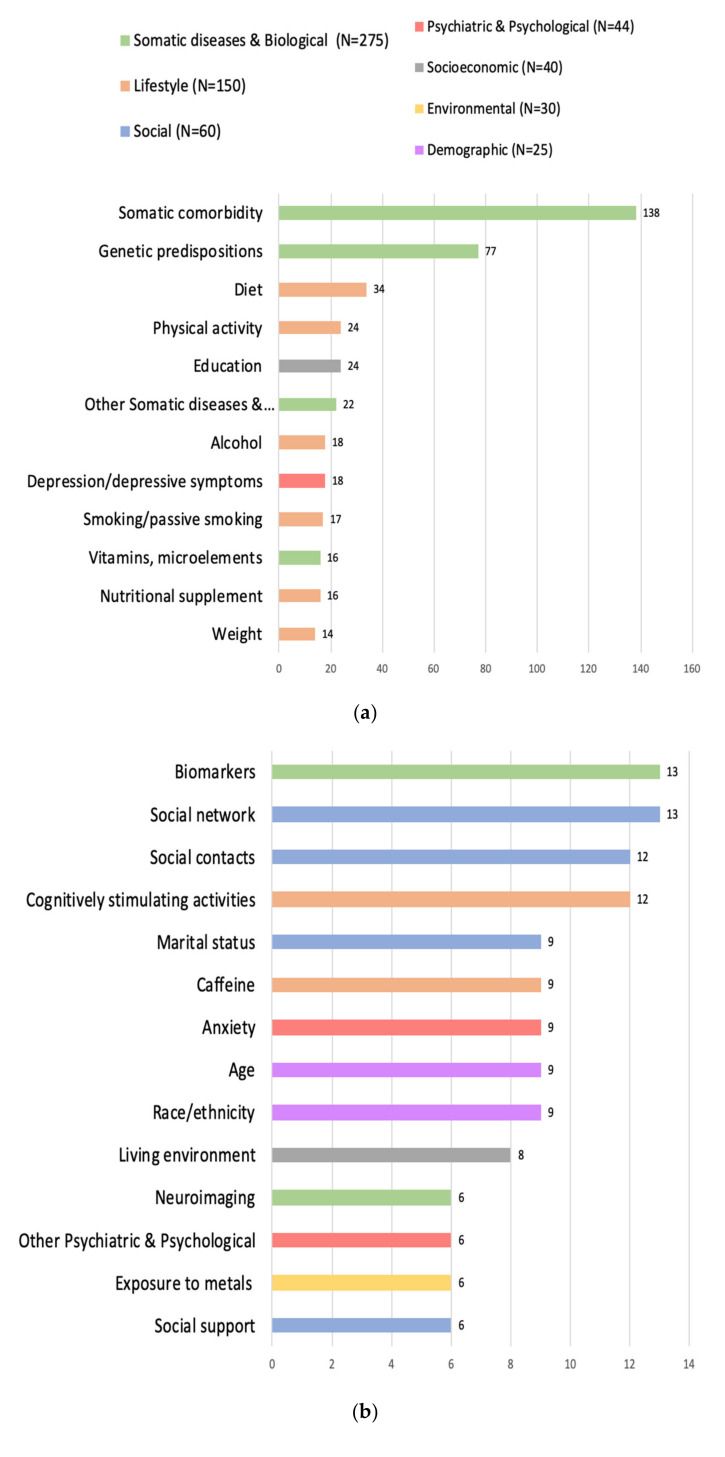
Frequency of reported factors grouped into subcategories (**a**–**c**).

**Figure 3 brainsci-12-01214-f003:**
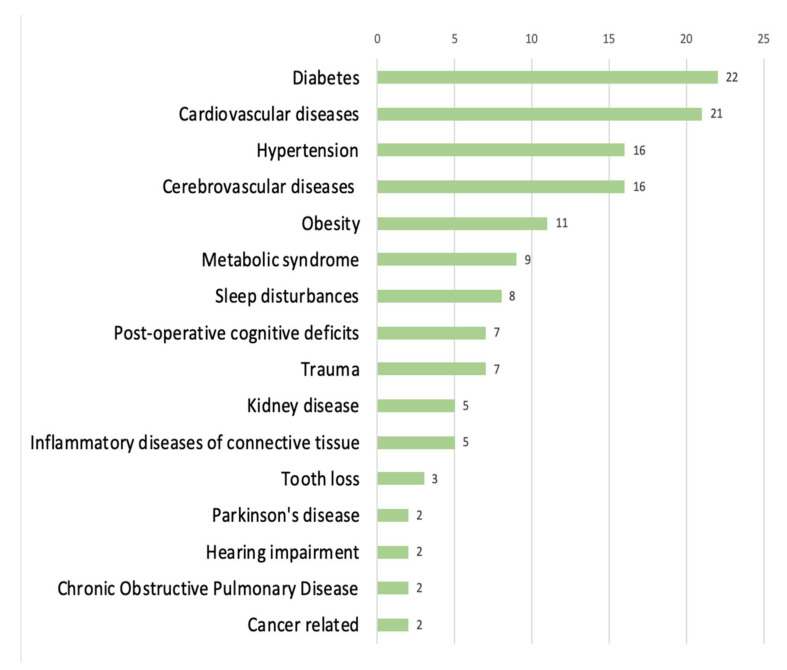
Frequency of reported factors grouped into the subcategory ‘somatic comorbidity’.

**Figure 4 brainsci-12-01214-f004:**
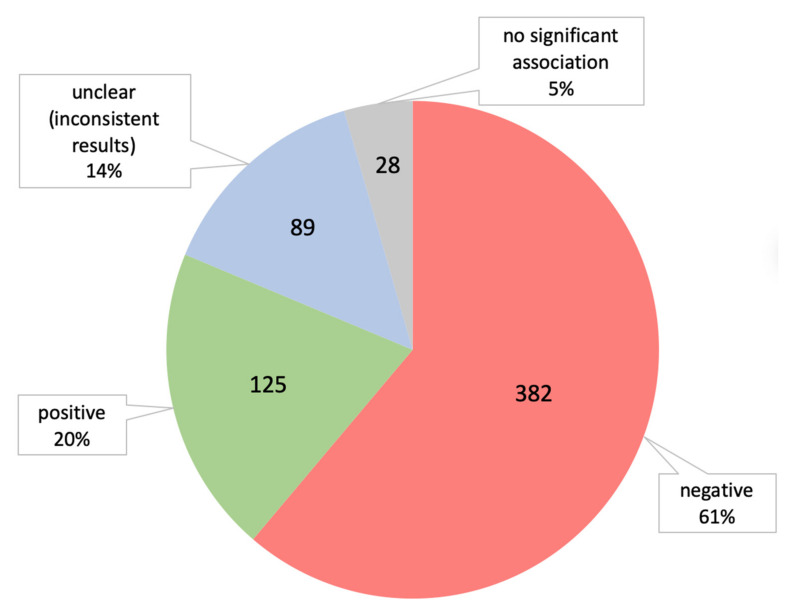
Type of association of factors with cognitive function.

**Figure 5 brainsci-12-01214-f005:**
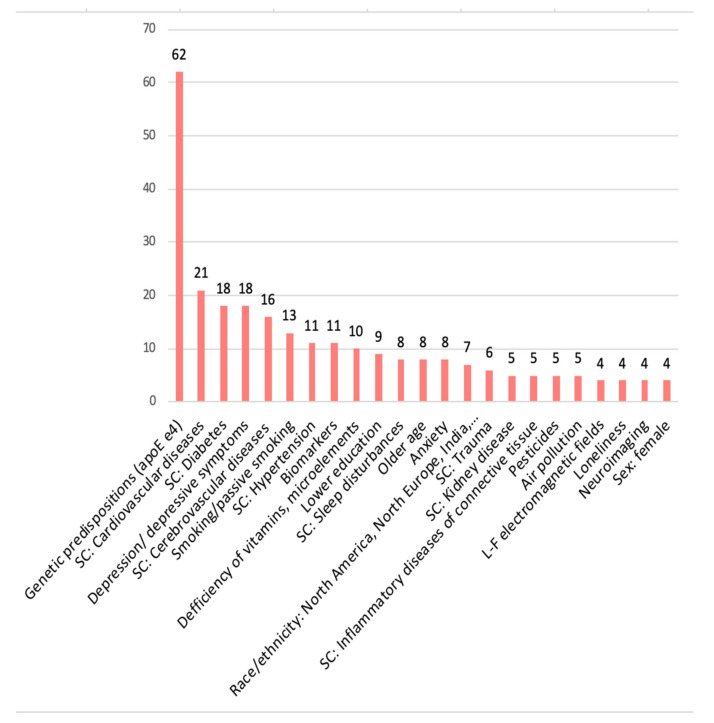
The most frequently studied risk factors.

**Figure 6 brainsci-12-01214-f006:**
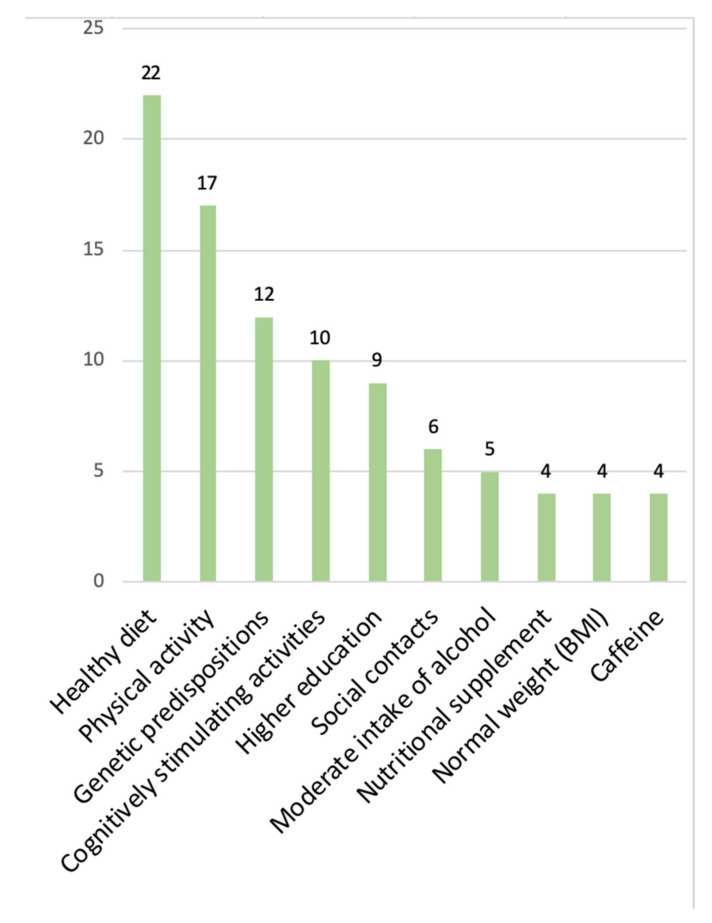
The most frequently studied protective factors.

**Figure 7 brainsci-12-01214-f007:**
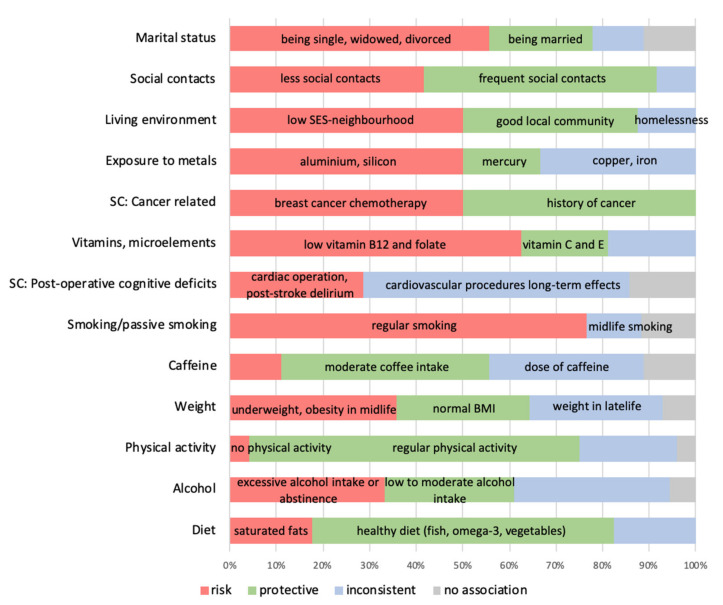
Factors with inconsistent or dual association with cognition.

**Figure 8 brainsci-12-01214-f008:**
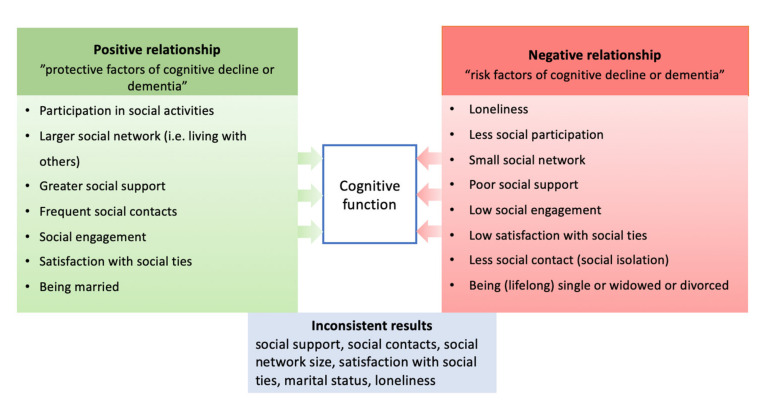
Social factors and their relationships with cognitive outcomes.

**Table 1 brainsci-12-01214-t001:** Characteristics of the studies included in the review (*n* = 314).

Type of Study	%	*n*
Systematic review (SR)	41.1	129
Meta-analysis (MA)	22.6	71
Systematic review and Meta-analysis	36.3	114
**Publication Year**		
2009	4.1	13
2010	3.5	11
2011	4.8	15
2012	7.3	23
2013	9.5	30
2014	7.6	24
2015	8.9	28
2016	13.4	42
2017	18.2	57
2018	13.1	41
2019	8.3	26
2020	1.0	3
2021	0.3	1
**Country (by first author)**		
European countries (UK)	35.8	112 (41)
China	28.3	89
USA	12.7	40
Australia	7.3	23
Canada	5.7	18
Asia (various)	5.4	17
South America	3.8	12
Africa	1.0	3
**Age of participants**		
Young adulthood and mid-life (<65)	0.9	3
Mid-life and late-life (≥40)	45.2	142
Late-life (≥65)	23.6	74
All ages (≥18)	13.7	43
No data	16.6	52
**Cognitive outcome variables (*n* = 462)**	%	*n* = 462
Cognitively Healthy (CH) [incl. Cognitive Reserve (CR)]	10.4	48 (2)
Subjective Cognitive Impairment (SCI)	0.6	3
Cognitive impairment/Cognitive decline (CI/CD)	24.0	111
Mild Cognitive Impairment (MCI)	10.4	48
Dementia (any type) (D) [incl. AD]	54.6 (34.6)	252 (160)

**Table 2 brainsci-12-01214-t002:** Final categories with the number of reported factors for each subcategory.

Category	Subcategory	Frequency of Reporting the Factors
Demographic (*n* = 25)	Age	9
	Race/ethnicity	9
	Sex	5
	Bilingualism	2
Environmental (*n* = 30)	Exposure to metals	6
	Air pollution	5
	Pesticides	5
	Sun exposure	5
	Other Environmental	4
	Low-frequency electromagnetic fields	4
Socioeconomic (*n* = 40)	Education	24
	Living environment	8
	Socioeconomic status (SES)	5
	Occupation	3
Psychiatric & Psychological (*n* = 44)	Depression/depressive symptoms	18
	Anxiety	9
	Other Psychiatric & Psychological	6
	Personality traits	3
	Subjective cognitive complaints	2
	Delirium	2
	Bipolar disorders	2
	Apathy	2
Social (*n* = 60)	Social network	13
	Social contacts/isolation	12
	Marital status	9
	Social support	6
	Participation in social activities	5
	Satisfaction with social ties	5
	Loneliness	5
	Social engagement	5
Lifestyle (*n* = 150)	Diet	34
	Physical activity	24
	Alcohol	18
	(passive) Smoking	17
	Nutritional supplement	16
	Weight	14
	Cognitively stimulating activities	12
	Caffeine	9
	Sleep pattern	3
	Other lifestyle	3
Somatic diseases & Biological (*n* = 275)	Somatic Comorbidity	138
	Genetic predispositions	77
	Other somatic diseases & biological	22
	Vitamins, microelements	16
	Biomarkers	13
	Neuroimaging	6
	Molecular factors	3

## Data Availability

Not applicable.

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
