# Peer review of "What Do We Know about Social and Non-Social Factors Influencing the Pathway from Cognitive Health to Dementia? A Systematic Review of Reviews"

_brainsci, 2022, doi:10.3390/brainsci12091214_

Round 1

Reviewer 1 Report

An interesting paper highlighting influential factors in the pathway from cognitive health to dementia. The authors carry out an exhaustive review of systematic reviews and meta-analyses analyzing 314 papers. 

It is a great work that can help scientific community. However, it brings little new to what is already known, so I propose modifications that can give value to the extensive work carried out. 

First of all, I'd list typographical errors and oversights: 

The supplementary material has the wrong numbering of the tables. When opening the Table S2 file, it is written Table 1 and so on with the rest of the files. 

In Table S4 it is missing to add in the last box: Other somatic comorbidity 

Line 183: 314 have to be written with letters and not numbers as they appear after a point. 

Line 216: 61 subcategories, but there are 62 

Line 236: Subgroupgenetic predispositionwas second in regard to the frequency of factors. It appears as the first in the figure. 

Line 242: The use of benzodiazepines would perhaps be better considered as lifestyle just like the use of cannabis. 

Line 246: replace subsets with subcategories 

In Figure 2: Adding the definition of SES: socioeconomic status 

Line 280: comma instead of point

Regarding the content and in order to take better advantage of the great systematic review carried out, I propose the following major modifications and improvements: 

Talking about 59 initially and 62 final subcategories can create confusion, also when trying to compare table S4 with figure 2, so it will be better to ignore the initial categories and therefore eliminate table S4. 

I propose to add a new table with the final categories and subcategories and the number of factors for each subcategory in the article and a supplementary material listing the factors for each subcategory in case they could not be listed in the new table. 

On the other hand, figure 3 is confused when talking about positive and negative associations and risk and protective factors. 

It is clear that somatic disease and biological categories are mostly risk factors, although there are categories that create confusion such as cancer that have 50% positive and 50% negative association. Could it be that studies have linked having cancer as a risk factor and not having it as a protector? The same happens with the rest of the factors that have negative and positive associations at the same time.

The weight subcategory, for example, does it refer to being overweight or normal or underweight? The marital status subcategory does not specify whether it refers to married, single, or widowed. Because being married has a positive association and is a protective factor, while being single or widowed is a risk factor. Isn't that the explanation for why the same factor is risky and protective? That is possible within the same subcategory the factor has been studied differently? 

This fact is clarified with Figure 4 in reference to social factors, so I propose to expand Figure 4 with the rest of the subcategories when duality appears and created confusion. 

And in relation to this proposed modification, broaden the discussion so that the subcategories that are risk factors and at the same time protectors can be understood. 

I believe that these suggestions can give greater value to the results presented in the work and that in this way the scientific community can be benefited. 

Reviewer 2 Report

It is a very interesting and appropriate review due to its relevant topic for the scientific and medical community. The methodology followed all the rigors of the guidelines, regarding the search, analysis, classification and detailing of inclusion and exclusion criteria. However, the following areas could be improved in the results session:

-The proportion of the cognitive outcome variable is missing from the results description of item 3.2.

-Standardize the decimal places for Table 1's percentage values.

-In Figure 2, the somatic comorbidity category left off the number of studies (N=?), the categories order is unclear, and the figure legend provides little to help explain what the figure information represents

-The findings of each category are described in the subitem of 3.4, which corresponds to the different color bars in Figure 2, however this explanation was left out from the text. As a result, there is a gap between the text and Figure 2's complexity/ details.

-In item 3.4.3 and 3.4.6 missing the number of social citations and in item 3.4.4 missing the percentage. 

- Figure 3 is shown with the size font very small and with low deffinition, this occured in the other figures too. 

- The entire description of Figure 3 in the sub-item of item 3.5 does not link to the details of this figure. It seems that the exposition of the results of the figure has no relevance.

Round 2

Reviewer 1 Report

Thank you very much for the effort put into the best article. It has improved a lot. One last question in reference to table 2:  What does number of factors refer to? I imagine that it does not refer to the different factors by category, because if it were, the sex category would be 2. If it refers to the frequency of reporting factors. Please change the column header to avoid confusion.

Author Response

Thank you very much for your valuable comment. Indeed, the numbers refer to the frequency of reporting the factors. We have changed the subheading in Table 2 accordingly.